# *bla*_NDM_ and *mcr-1* to *mcr-5* Gene Distribution Characteristics in Gut Specimens from Different Regions of China

**DOI:** 10.3390/antibiotics10030233

**Published:** 2021-02-25

**Authors:** Dongyue Lv, Ran Duan, Rong Fan, Hui Mu, Junrong Liang, Meng Xiao, Zhaokai He, Shuai Qin, Jinchuan Yang, Huaiqi Jing, Zhaoguo Wang, Xin Wang

**Affiliations:** 1Department of Epidemiology and Health Statistics, The School of Public Health of Qingdao University, Qingdao 266021, China; lvdongyue0707@163.com; 2State Key Laboratory of Infectious Disease Prevention and Control, National Institute for Communicable Disease Control and Prevention, Chinese Center for Disease Control and Prevention, Beijing 102206, China; duanran@icdc.cn (R.D.); fr247616716@126.com (R.F.); mh101457@163.com (H.M.); liangjunrong@icdc.cn (J.L.); xiaomeng@icdc.cn (M.X.); hezhaokai1995@163.com (Z.H.); qinshuai@icdc.cn (S.Q.); xzyjc@126.com (J.Y.); jinghuaiqi@icdc.cn (H.J.)

**Keywords:** *bla*_NDM_, *mcr*, carbapenem, polymyxin

## Abstract

Antibiotic resistance has become a global public health concern. To determine the distribution characteristics of *mcr* and *bla*_NDM_ in China, gene screening was conducted directly from gut specimens sourced from livestock and poultry, poultry environments, human diarrhea patients, and wild animals from 10 regions, between 2010–2020. The positive rate was 5.09% (356/6991) for *mcr* and 0.41% (29/6991) for *bla*_NDM_, as detected in gut specimens from seven regions, throughout 2010 to 2019, but not detected in 2020. The detection rate of *mcr* showed significant differences among various sources: livestock and poultry (14.81%) > diarrhea patients (1.43%) > wild animals (0.36%). The detection rate of *bla*_NDM_ was also higher in livestock and poultry (0.88%) than in diarrhea patients (0.17%), and this was undetected in wildlife. This is consistent with the relatively high detection rate of multiple *mcr* genotypes in livestock and poultry. All instances of coexistence of the *mcr-1* and *bla*_NDM_ genes, as well as coexistence of *mcr* genotypes within single specimens, and most new *mcr* subtypes came from livestock, and poultry environments. Our study indicates that the emergence of *mcr* and *bla*_NDM_ genes in China is closely related to the selective pressure of carbapenem and polymyxin. The gene-based strategy is proposed to identify more resistance genes of concern, possibly providing guidance for the prevention and control of antimicrobial resistance dissemination.

## 1. Introduction

Antibiotic resistance has become a major global public health concern in the 21st century. Carbapenems and polymyxins are among the last-resort antibiotics for defending against Gram-negative bacterial infections [1]. Among the various mechanisms, the *bla*_NDM_ (New Delhi metallo-β-lactamase) gene and the *mcr* (mobile colistin resistance) gene, conferring resistance to carbapenem and polymyxin respectively, exhibit cross-species and cross-region transmission [2]. The *bla*_NDM-1_ gene was first discovered in patients in 2009 [3], and 29 genotypes have since been identified [4,5,6,7,8,9]. Carbapenems are mainly used to treat human respiratory infections, and resistant bacterial strains often exhibit multidrug and broad-spectrum drug resistance [10]. Although carbapenem usage has not been approved for use in the breeding industry in China, new *bla*_NDM_ subtypes and an epidemic of resistant bacterial strains have appeared in livestock and poultry [11,12]. The *mcr-1* gene was first discovered in pigs in 2016 [13] and 10 genotypes and multiple subtypes have since been found [14,15,16,17,18]. Polymyxins were once widely used as feed additives and for disease prevention in livestock and poultry in China, but have been banned as animal growth promoters in China since 2017. The colistin-resistant *Escherichia coli* (CREC) and *mcr-1* positive *Escherichia coli* (MCRPEC) seem to have declined [19], however, our results from multiple sources did not show a decrease [20]. The *mcr*-positive or polymyxin-resistant strains are not only found in livestock, humans, and environments [21,22,23], but also in wild animals such as macaques and migratory birds [24,25,26].

Antibiotic overuse [27] and the emergence of drug resistance are linked [28]. China has become the largest consumer of antibiotics due to antibiotic use in the livestock and poultry industries [29]. Our previous work showed that most of the *mcr* or *bla*_NDM_ positive strains were from normal flora, identified in isolates from wildlife, patients, livestock and poultry, and environmental specimens [20], with these appearing to be increased by antibiotic selective pressures. However, normal flora with resistance may not always be detected in this way due to the limited sensitivity of isolation techniques. This study was further conducted to determine the broader picture of *mcr* and *bla*_NDM_ in China, in the context of these genes being carried by bacteria selected by antibiotic pressures and the normal flora with resistance, providing guidance for drug resistance control measures.

## 2. Results

### 2.1. Distributions of bla_NDM_ and mcr

Among the 6991 gut specimens collected, 0.41% were positive for *bla*_NDM_ (29/6991) and 5.09% were positive for *mcr* (356/6991) (Table 1, detailed background in Appendix A). For the *bla*_NDM_ gene, the detection rate of *bla*_NDM-1_ (0.37%, 26/6991) was the highest, followed by *bla*_NDM-24_ (0.04%, 3/6991). For the *mcr* gene, the detection rate of *mcr-1* (4.79%, 335/6991) was highest, followed by *mcr-2* (0.29%, 20/6991), *mcr-3* (0.16%, 11/6991), and *mcr-4* (0.11%, 8/6991). No gut specimen was positive for *mcr-5*.

The majority of *bla*_NDM_ and *mcr* genes were found in livestock and poultry. The positive rates of the *bla*_NDM_ gene in livestock and poultry, poultry environments, and diarrhea patients were 0.88 (16/1823), 3.14 (11/350), and 0.17% (2/1186), respectively; it was not detected in wild animals. The *bla*_NDM_ gene rates between various sources showed significant differences (Fisher exact test χ^2^ = 22.66, *p* < 0.05). The positive rates of the *mcr* gene in livestock and poultry, poultry environments, diarrhea patients, and wild animals were 14.81 (270/1823), 16.00 (56/350), 1.43 (17/1186), and 0.36% (13/3632), respectively. The *mcr* gene rates between various sources showed significant differences (Pearson χ^2^ = 643.72, *p* < 0.05). The detection rate of *bla*_NDM_ and *mcr* within single gut specimens was 0.38 (7/1823) and 0.86% (3/350) for livestock and poultry, and poultry environments, respectively. Among the positive gut specimens from livestock and poultry, intensively reared animals (swine, chickens and fish) accounted for a greater proportion than non-intensively reared breeding animals (yak, goats and canines) (Figure 1). The positive rates of these two kinds of reared animals were 1.02 (12/1179) and 0.62% (4/644) for the *bla*_NDM_ gene, 21.80 (257/1179) and 2.02% (13/644) for the *mcr* gene. In wild animals, the *mcr* gene was detected in various species ranging from marmots and rats to bats (Figure 1).

Regarding gut specimen collection year, *bla*_NDM_-positive gut specimens were detected in 2011 (0.27%, 1/372), 2017 (0.56%, 2/356), 2018 (0.28%, 5/1797), and 2019 (1.33%, 21/1575), and *mcr*-positive gut specimens were detected throughout 2010–2019, with detection rates of 1.80 (9/501), 5.38 (20/372), 1.01 (9/889), 5.33 (17/319), 22.95 (95/414), 1.41 (5/354), 0.67 (2/299), 0.84 (3/356), 1.61 (29/1797), and 10.60% (167/1575). More specifically, *bla*_NDM-1_ was found in 2011, 2017, 2018, and 2019, *bla*_NDM-24_ was found in 2019, *mcr-1* was found throughout 2010–2019 (with detection rates peaking in 2014 and 2019), *mcr-2* was found in 2014, *mcr-3* was found in 2010, 2012, and 2019, and *mcr-4* was found in 2011, 2014, 2015, and 2019. New *mcr* subtypes were found in 2010 (*mcr-3.32*), 2012 (*mcr-3.31*), 2014 (*mcr-1.30*, *mcr-2.4*, *mcr-2.5*, *mcr-2.6*, *mcr-2.7*), and 2018 (*mcr-1.29*) (Figure 2).

### 2.2. Sequence Analysis of bla_NDM_ and mcr

Among the *bla*_NDM_-positive gut specimens, no new mutations were found. Among these gut specimens, 89.66% (26/29) were identical to NDM-1 (Accession No.: WP_004201164.1) and 10.34% (3/29) were identical to NDM-24 (Accession No.: WP_111672913.1). The amino acid (aa) identity between NDM-1 and NDM-24 was 99.8%. Among the *mcr*-positive gut specimens, 8.15% (29/356) involved new subtypes of *mcr-1* to *mcr-3*. No new mutations were found in *mcr-4*, which all belonged to MCR-4.3 (Accession No.: WP_011638903.1). A cluster analysis of the *mcr* genotypes is shown in Figure 3.

Among the *mcr-1*-positive gut specimens, 98.21% (329/335) were identical to MCR-1.1 (Accession No.: WP_049589868.1). The remaining six gut specimens formed two new subtypes, all with a sense mutation (Figure 3). The aa identities of each of the new subtypes compared with MCR-1.1 were: 99.8 (MCR-1.29) and 99.8% (MCR-1.30). The amino acid mutation of MCR-1.29 is P397S, and MCR-1.30 is G474D.

All 20 *mcr-2* positive gut specimens in this study were new subtypes. Compared to MCR-2.1 (Accession No.: WP_065419574.1), there were numerous sense and nonsense mutations (Figure 3). Compared with MCR-2.1, the aa identity of MCR-2.4 was 97.0%, and of the 3% mutations, 81.81% were nonsense and 18.18% were sense; the aa identity of MCR-2.5 was 98.5%, and of the 1.5% mutations, 84.91% were nonsense and 15.09% were sense; the aa identity of MCR-2.6 was 98.7%, and of the 1.3% mutations, 83.39% were nonsense and 10.61% were sense; the aa identity of MCR-2.7 was 98.5%, and of the 1.5% mutations, 85.96% were nonsense and 14.04% were sense.

Among the 11 *mcr-3* positive gut specimens, two were consistent with MCR-3.18 (Accession No.: WP_111273847.1) and two were identical to MCR-3.3 (Accession No.: WP_099982814.1). The remaining seven gut specimens formed two new subtypes, both involving the premature stop codon that was one codon before the expected stop codon (Figure 3). The aa identities of each of the new subtype compared with MCR-3.1 (Accession No.: WP_039026394.1) were: 94.4% (MCR-3.31, sense mutation: 41.03%, nonsense mutation: 58.97%) and 94.6% (MCR-3.32, sense mutation: 40.26%, nonsense mutation: 59.74%). Both MCR-3.31 and MCR-3.32 had a premature stop codon at aa 541 out of 542.

### 2.3. Distribution of Coexisting Genes/Genotypes and New Subtypes

The gut specimens with coexisting genes/genotypes or new subtypes were mostly from livestock and poultry and poultry environment, with only the new subtype *mcr-3.31* being derived from wild animals (Table 2). Ten gut specimens harbored both *bla*_NDM_ and *mcr-1*, and 18 gut specimens harbored two *mcr* genotypes. The coexistence of *bla*_NDM_ and *mcr-1* within a single gut specimen was only observed in Anhui. The coexistence of *mcr* genotypes a within single gut specimen was mostly found in Guangxi. The new *mcr* subtypes were from Guangxi, Anhui and Yunnan.

## 3. Discussion

Antibiotic resistance may be a survival strategy for bacteria, with antibiotics triggering specific bacterial responses [30,31]. This study shows that antibiotic selective pressure might be reflected by resistance gene pools of various sources. In combination with the findings of our previous study, this shows that the emergence of polymyxin and carbapenem resistance strains in China is closely related to the selective pressure of antibiotics. The *mcr* or *bla*_NDM_ strains originating from livestock and poultry, patients, and wildlife, are mainly non-pathogenic organisms [20], which is consistent with findings from studied conducted in 47 countries across six continents with *mcr*-positive strains [32], which showed a tendency to be increased under antibiotic selection pressures. Due to the limited sensitivity of isolation, some normal flora with resistance may not be detected. To further determine the distribution of *mcr* and *bla*_NDM_ from different sources—carried by the bacteria selected by antibiotic selective pressures and normal flora with resistance—this study was conducted based on gut specimen detection strategy and a One Health approach. Overall, the positive rate of the *mcr* gene was much higher than that of the *bla*_NDM_ gene for each specimen source. This is in accordance with positive-strain isolation [20]. The positive rates of the *mcr* gene showed significant differences among sources: livestock and poultry (14.81%) > diarrhea patients (1.43%) > wild animals (0.36%) (Table 1), consistent with the relative isolation rates of polymyxin-resistant strains among these sources [20]. Though polymyxin-resistant strains had not been isolated in wildlife, the *mcr* gene was detected. Livestock and poultry (0.88%) were found to contain the *bla*_NDM_ gene more frequently than diarrhea patients (0.17%), but this gene was not detected in wildlife (Table 1). Carbapenem-resistant strains were also not isolated from wildlife in a previous study [20]. Compared with other sources, no polymyxin- or carbapenem-resistant strains [20], lower rates of *mcr* and *bla*_NDM_ genes (Table 1) and less *mcr* genotypes (Table 2) were found in wildlife samples, which supports the hypothesis that wild animals are a net sink rather than a source of clinically relevant drug resistance [33]. The phenotypic diversity of drug resistant strains in wildlife is also low [33]. Since wild animals have less chance of being exposed to antibiotics, the emergence of resistance genes possibly reflects the resistance genes carried by normal flora. Similarly, *Salmonella enterica*—isolated from diarrhea patients and asymptomatic individuals—showed equal carriage of *mcr* carriers, suggesting the *mcr* gene is carried by normal flora [34]. In this study, the detection rates of *mcr* and *bla*_NDM_ in diarrhea patients were far lower than in livestock and poultry, and higher than in wild animals (Table 2). This is in accordance with the relatively low use of polymyxin and carbapenem in this population.

The gene pools of *mcr* or *bla*_NDM_ reflect resistance genes carried by normal flora when antibiotic pressure is low, and genes carried by the bacteria selected by antibiotic pressure. When the pressure is relatively high, such as in livestock, poultry and humans, the relative levels of the *mcr* and *bla*_NDM_ genes—to a certain extent—possibly reflects the antibiotic selective pressure. In particular, in livestock and poultry, there higher rates of the *bla*_NDM_ and *mcr* gene (Table 1) and more *mcr* genotypes were found (Table 2 and Figure 3). Polymyxins are often used as therapeutic drugs and feed additives for animals, and they are used more frequently for farmed animals in China [29], where the highest number of *mcr*-positive strains was reported [32]. During the intensive feeding period, antibiotics are required for animal treatment and disease prevention, which involves large doses and long-term use [35]. We found that the positive rate of the *mcr* gene was much higher in intensively reared animals (21.80%, 257/1179, swine, chickens, etc.) than in non-intensively reared breeding animals (2.02%, 13/644, yak, goats, canine, etc.) (Figure 1). The ban of polymyxin use as an animal growth promoter in 2017 seems to have reduced CREC and MCRPEC [20,36]. However, the observation that the *mcr* detection rates peaked in 2019 in this study (10.60%, 167/1575) (Figure 2), is consistent with the notion that *mcr-1* isolates successively recovered from 2017 to 2019, which indicates the possibility that polymyxin resistance still exists in livestock and poultry. Carbapenem drug-resistant strains have appeared and are prevalent in poultry and livestock. New genotypes of the *bla*_NDM_ gene have been found in livestock and poultry-derived strains around the world [11,28]. Firstly, carbapenems might be applied when treating animal diseases. Secondly, their use in humans pollutes the environment and results in indirect exposure of animals to the drug. Last but not least, bacteria with the *bla*_NDM_ gene may exist in normal gut flora [37]. In summary, the emergence of drug resistance genes is due to the selective pressure caused by the overuse of antibiotics. The strategy of gene detection can be used for resistance gene profiles and supervision.

In this study, livestock and poultry were not only the main source of the *mcr* and *bla*_NDM_ gene pool (Table 1), but they were also sources of *mcr-1* and *bla*_NDM_ co-harbored genes. Additionally, livestock and poultry were the source of multiple *mcr* genotypes within single gut specimens (Table 2). Similar findings were not shown in diarrhea patients or wild animals. In general, the coexistence of the *mcr-1* and *bla*_NDM_ genes was only found in Anhui, and the coexistence of *mcr* genotypes mostly came from Guangxi, indicating that livestock and poultry in some regions may be exposed to higher or more complex antibiotic selective pressures. Considering that no strain carrying both the *mcr* and *bla*_NDM_ genes had been isolated in the previous study [20], a past and present coexistence of the *mcr-1* and *bla*_NDM_ genes within one gut specimen is more likely to come from different clones (e.g., one clone harboring *mcr-1*, other clone harboring *bla*_NDM_). It is also possible that a single clone carried both genes. In either case, the drug resistance conferred by *mcr* and *bla*_NDM_ genes may be transmitted from livestock and poultry to humans, possibly even resulting in the emergence of polymyxin and carbapenem resistant strains. Recently, *mcr-1* and *bla*_NDM_ coexistence was also reported in the United States, Venezuela, and Japan [38,39,40], which reduces treatment options for multidrug-resistant bacterial infections and increases the incidence and mortality of the infections, leading to stricter antibiotic controls. It is necessary to strengthen antimicrobial resistance surveillance in livestock and poultry.

This study revealed the gene distribution of *mcr* and *bla*_NDM_ in livestock and poultry, diarrhea patients and wild animals, demonstrating that relative level of the resistance genes may reflect the selective pressure of antibiotic exposure of various hosts, which is expected to become a strategy of antibiotic usage oversight. Potential antimicrobial usage of colistin, and others, plays a role in the enrichment of antimicrobial resistance genes in gut specimens, which are needed to further support culture-based data. Compared with the culture-based strategy, the gene-based strategy is more sensitive. The positive rates of gene detection among various gut specimens were about two to three times those of isolation rates [20]. On the other hand, bacterial culture and genetic background information is not available through the gene-based strategy. The fact that more positive specimens found by gene detection than culture detection, may come from the normal flora with resistance that cannot always be isolated, and gene positive results do not always equate to phenotype positive results [25]. Additionally, searching for new variants is limited by the current PCR method. Although this method is improving over time [34,41], it is based on known genotype data which often cannot be used to discover an unknown variant. The gene detection method could be developed into a strategy based on metagenomic sequencing [42], identifying more concerned drug resistance genes and genetic information coming from various sources, and providing guidance for the prevention and control of drug-resistant bacteria and for supervision of antibiotic usage.

## 4. Materials and Methods

### 4.1. Gut Specimen Sources

Nucleic acid samples were obtained from 6991 gut specimens from livestock and poultry (26.08%, 1823/6991) including swine, chickens, canine, yak, goats, etc., poultry environments (5.01%, 350/6991), including breeding or slaughter environment, human diarrhea patients (16.96%, 1186/6991), and wild animals (51.95%, 3632/6991), including bats, marmots, rats, etc. (Table 1). The gut specimens were obtained in 2010–2020 from 10 regions of China (Beijing, Anhui, Gansu, Yunnan, Guangxi, Guizhou, Ningxia, Inner Mongolia, Qinghai, and Zhejiang) (Appendix A), and they were retrospectively screened for the target genes. Gut specimen types of this study included human feces, animal anal swab, feces, intestinal content/swab or oral-pharyngeal swab, and poultry environment specimens related to gut environment, including drinking water, cage swab, depilator swab, cleaning sewage, chopping board swab, and soil. Unified protocols for specimen collection, transportation, and process were applied by professionals from local CDC (Center for Disease Control and Prevention) facilities, Institutes for Endemic Disease Prevention and Control, and hospitals. Specimens were collected and transported in Cary–Blair Transport Medium, processed and nucleic acids extracted using a genomic extraction kit (TIANamp Bacteria DNA Kit, Beijing, China). The nucleic acid samples were frozen for storage.

### 4.2. bla_NDM_ and mcr-1 to mcr-5 Screening of Gut Specimens and Sequence Analysis

The target genes *bla*_NDM_ and *mcr* were screened for, sequenced, and aligned with reference sequences from the National Center for Biotechnology Information (NCBI) database. The screening primers (Appendix A) for *bla*_NDM_ and *mcr* (*mcr-1* to *mcr-5*) were previously described [20,41]. The original amplification of *mcr-1* to *mcr-5* involved multiplex PCR, but single PCR was conducted in this study. The CDS (coding sequences) of gut specimens with mutations in the screening sequences were further amplified, cloned (Transgene, Beijing, China), and sequenced. The number of PCR cycles for gene screening is 25 to 30, for CDS amplification it is 30. The PCR was performed using a 20 μL volume containing 10 μL Premix Taq version 2.0 (Takara, Beijing, China), 8 μL ultrapure distilled water, 0.5 μL (10 μM) of each forward and reverse primer and 1 μL of DNA template. The amplified products were detected using gel electrophoresis and sequenced in both directions using an Applied Biosystems 3730xl DNA Analyzer (Tsingke Biological Technology, Beijing, China). Phylogenic tree was constructed based on CDS sequences of *mcr* gene including sequences of this study and reference sequences (*mcr-1* to *mcr-4*) and sequence analysis of *mcr* and MCR were conducted (Figure 3).

### 4.3. Statistical Analysis

Pearson’s chi-square test (theoretical frequency *T* ≥ 5) was used to compare positive rates among different sources. As one theoretical frequency is 1 < *T* < 5, the Fisher exact test was also applied when comparing rates among different sources. Bonferroni correction was used to compare the positive rates between two sources. *p* < 0.05 was considered statistically significant. The statistical analysis was conducted by SPSS Version 19.0.

### 4.4. Nucleotide Sequence Accession Numbers

The CDSs of the following new subtypes (*mcr-1* to *mcr-3*) were deposited in the GenBank database: *mcr-1.29* (GenBank: MT731964), *mcr-1.30* (GenBank: MT731965), *mcr-2.4* (GenBank: MT757845), *mcr-2.5* (GenBank: MT757842), *mcr-2.6* (GenBank: MT757844), *mcr-2.7* (GenBank: MT757843), *mcr-3.31* (GenBank: MT757846), and *mcr-3.32* (GenBank: MT757847).

### 4.5. Ethics Statement

The study was approved by the ethics committee of the National Institute for Communicable Disease Control and Prevention of the Chinese Center for Disease Control and Prevention (IACUC Issue No. 2020-008). Verbal consent was obtained from the included diarrhea patients.

## 5. Conclusions

This study is first to determine the distribution characteristics of *bla_NDM_* and *mcr* genes from various sources of China. The positive rate of the *mcr* gene was much higher than that of the *bla*_NDM_ gene for all sources, from highest to lowest was: livestock and poultry, diarrhea patients, and wild animals. The *mcr* or *bla*_NDM_ gene pool of certain source reflect the resistance gene carried by normal flora when antibiotic pressure is low, and genes carried by the bacteria selected by antibiotic pressure. Livestock and poultry were not only the main source of the *mcr* and *bla*_NDM_ gene pool, but also the source of co-harbored *mcr-1* and *bla*_NDM_ genes. The antimicrobial resistance surveillance in livestock and poultry needs to be strengthened. In conclusion, the study demonstrated that the selective pressure of antibiotic exposure of various hosts maybe reflected by relative level of the resistance genes, which is expected to become a strategy of antibiotic usage oversight.

## Figures and Tables

**Figure 1 antibiotics-10-00233-f001:**
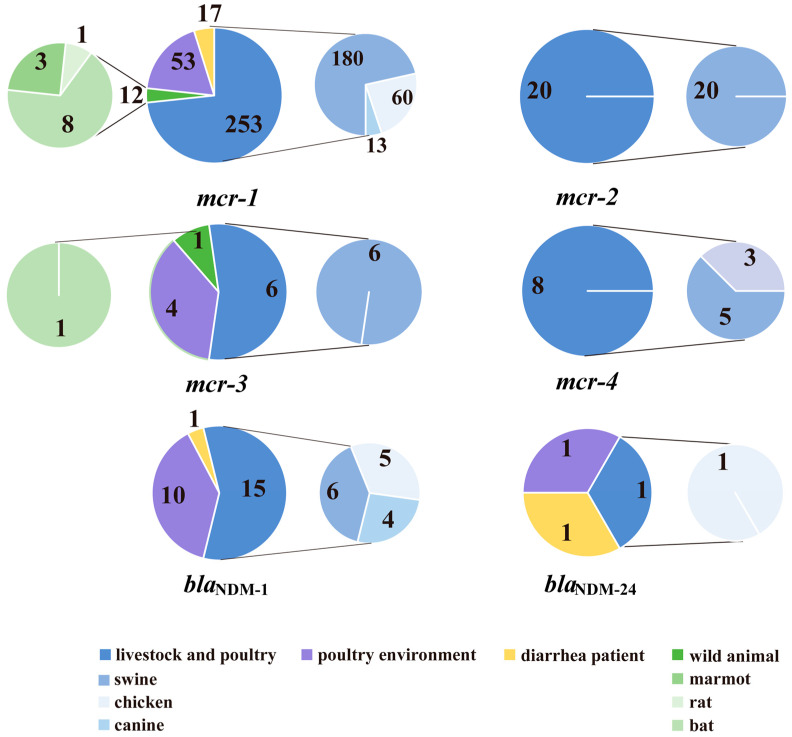
Gut specimen sources of each *bla*_NDM_ and *mcr* subtype.

**Figure 2 antibiotics-10-00233-f002:**
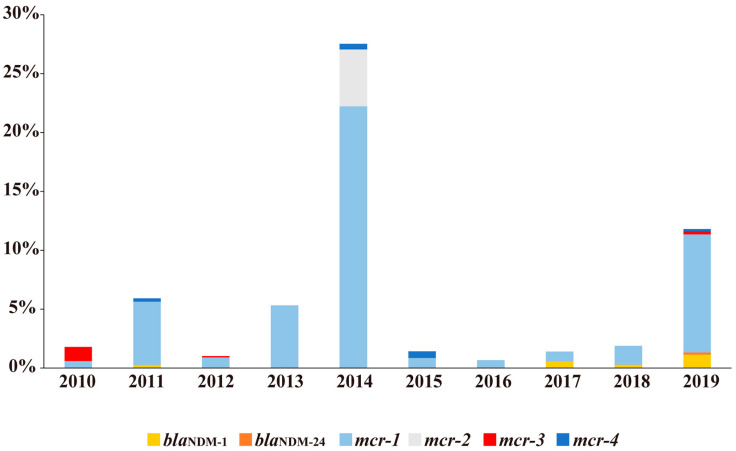
Positive detection rates of *mcr* or *bla*_NDM_ gene from 2010 to 2019. Neither was detected in 2020.

**Figure 3 antibiotics-10-00233-f003:**
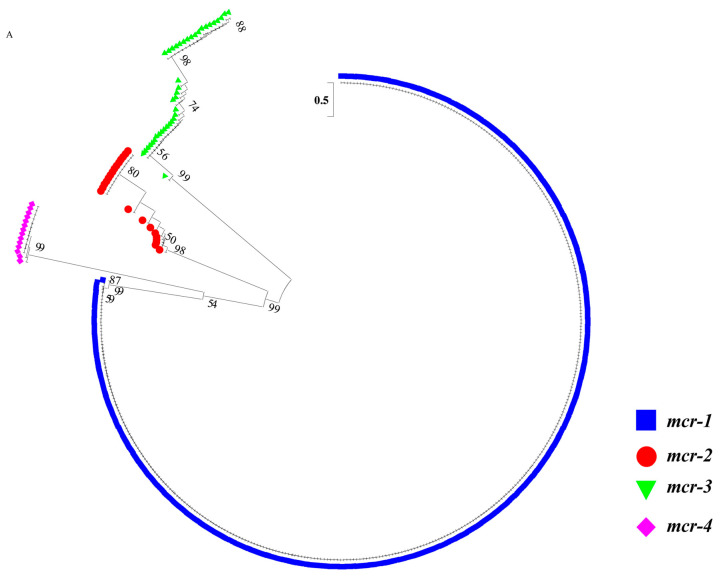
Cluster analysis of *mcr* genotypes (**A**) and nucleotide (nt) and amino acid (aa) alignments of new *mcr* subtypes (**B**,**C**). (**A**) Phylogenic tree of *mcr-1* to *mcr-4* based on nt sequences. (continued). (**B**) nt mutations of new *mcr* subtypes. Sense mutation is shown in yellow, nonsense mutation is shown in red. (**C**) aa mutations of new *mcr* subtypes. Sense mutation is shown in yellow, nonsense mutation is shown in red. * termination codon.

**Table 1 antibiotics-10-00233-t001:** Positive rate of *bla*_NDM_ and *mcr* in gut specimens from various sources.

Source	No. Specimens	*mcr* (%) *	*bla*_NDM_ (%) *	*mcr* and *bla*_NDM_ (%)
Livestock and poultry	1823	14.81 _a_	0.88 _a_	0.38
Poultry environments	350	16.00 _a_	3.14 _b_	0.86
Diarrhea patients	1186	1.43 _b_	0.17 _c_	-
Wild animals	3632	0.36 _c_	-	-
Total	6991	5.09	0.41	0.14

* The positive rate shows significant differences between different sources (*p* < 0.05). _a_, _b_, _c_: each subscript letter denotes a subset of source categories whose column proportions do not differ significantly from each other.

**Table 2 antibiotics-10-00233-t002:** Distribution of genotypes and coexisting genes/genotypes.

Gene	Genotype	Livestock and Poultry	Poultry Environments	Diarrhea Patient	Wild Animals	Total
*bla*_NDM_ or *mcr*	*bla* _NDM-1_	9	8	1		18
	*bla* _NDM-24_			1		1
	*mcr-1.1*	228	48	17	12	305
	*mcr-1.29 **		1			1
	*mcr-1.30 **	1				1
	*mcr-2.4 **	3				3
	*mcr-2.6 **	1				1
	*mcr-3.18*		2			2
	*mcr-3.3*		1			1
	*mcr-3.31 **				1	1
	*mcr-3.32 **	6				6
	*mcr-4.3*	7				7
	*mcr-1.1, mcr-2.4 **	2				2
	*mcr-1.1, mcr-2.5 **	1				1
	*mcr-1.1, mcr-2.7 **	9				9
	*mcr-1.1, mcr-3.3*		1			1
	*mcr-1.1, mcr-4.3*	1				1
	*mcr-1.30 *, mcr-2.7 **	4				4
*bla*_NDM_ and *mcr*	*bla*_NDM-1_, *mcr-1.1*	6	2			8
	*bla*_NDM-24_, *mcr-1.1*	1	1			2
Total		279	64	19	13	375

* new subtypes found in this study.

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
