# Peer review of "blaNDM and mcr-1 to mcr-5 Gene Distribution Characteristics in Gut Specimens from Different Regions of China"

_antibiotics, 2021, doi:10.3390/antibiotics10030233_

Round 1
Reviewer 1 Report
Please see attachment.

Author Response
Response to Reviewer 1 Comments
Point 1: - the manuscript reports on a very relevant subject, the resistance to antibiotics. This is a primordial health issue and the studies conducted within this subject are very important. However, different issues should be addressed prior to publication and I would advise against publication (rejection) given several key informations are missing on the materials and methods section and the English from the manuscript must be completely revised prior to any publication. The authors state in some places that the study was conducted between 2010-2020 and in other places between 2010-2019.
Harmonize.
Response 1: We greatly appreciate the reviewer’s professional comments and suggestions on our manuscript, with which it may have been much improved. We have studied all the comments carefully and made revisions accordingly. Several key informations are added to the materials and methods.
In this study, all the 6,991 specimens were collected between 2010-2020, which were screened for target genes. As a result, positive specimens with targets detected were collected between 2010-2019. We have checked through the manuscript to revise inappropriate and to avoid misleading description. Accordingly, we have revised it in abstract (line 17) and the title of Figure 2 was revised as, “Positive detection rates of mcr or blaNDM gene from 2010 to 2019. Neither was detected in 2020.” on line 105 to 106.
We are deeply sorry for the language problem caused by our writing and we have tried to fix them during revision. We have also invited professor Xuejie Yu for language editing this time, we have been in good partnership all along. Professor Yu is a Chinese American, who is the editors of many English-language journals. One of his latest publications in MDPI is: Pathogens. 2021 Jan 23;10(2):115. doi: 10.3390/pathogens10020115.
Point 2: Introduction:
- line 34: remove the word ‘resistance’
- line 43: write Escherichia coli prior to using the abbreviation
Response 2: Thanks for the comments. According to the comments, the word ‘resistance’ has been removed. The full name of E. coli has been added before abbreviation as, “colistin-resistant Escherichia coli (CREC) and mcr-1 positive Escherichia coli (MCRPEC)” on line 43 to 44.
Point 3: Results:
- table 1 - what are poultry environments? This is also not described in the materials and methods section. Additionally, it is not described in the materials and methods section what types of samples were collected from all animals, this must be described;
Response 3: Thanks for the comments. In this study, poultry environments refer to breeding or slaughter environment. The specimen types were added to methods section as, “... animal anal swab, feces, intes tinal content/swab or oral-pharyngeal swab, and poultry environment specimens related to gut environment, including drinking water, cage swab, depilator swab, cleaning sewage, chopping board swab, and soil.” on line 263 to 266.
Point 4: Results:
- line 72 - the authors state that the majority of the resistance genes were found in livestock and poultry. Given this is a subset of the sources analysed and that poultry environment, whatever this is, has the highest % of detection for both genes, did the authors include this as ‘livestock and poultry’? They always keep all the sources separated and if they are, then this sentence is incorrect;
Response 4: Thanks for the comments and sorry for the misleading description here. Yes, ‘livestock and poultry’ and ‘poultry environment’ are two different subsets of the sources through the manuscript, which is the same case here. According to Table 1, the highest rates of both genes came from poultry environment. But when comparing the total number of specimens, ‘livestock and poultry’ is higher than ‘poultry environment’. For mcr-positive samples, it is 14.81% (270/1823) for the former and 16.00% (56/350) for the latter one. For blaNDM-positive samples, it is 0.88% (16/1823) for the former and 3.14% (11/350) for the latter. Hence, it is summarized “The majority of blaNDM and mcr genes were found in livestock and poultry”, where Table 1 is inappropriate to be cited here. Accordingly, “(Table 1)” is removed on line 73.
Point 5: Results:
- why have the authors used two different statistical tests to evaluate the differences between samples;
Response 5: Thanks for the comments. Two statistical tests were applied in this study, according to different conditions. When comparing multiple sample rates (RxC table), Pearson's chi-square test can be used when conditions meet: number of samples n≥40, as well as theoretical frequency T≥5. Fisher exact test was applied under either situation: more than 1/5 theoretical frequency is 1<T<5 or T<1. Accordingly, it has been revised as, “Pearson's chi-square test (theoretical frequency T≥5) was used to compare positive rates among different sources. As one theoretical frequency is 1<T<5, Fisher exact test was also applied when comparing rates among different sources.” on line 289 to 291.
Point 6: Results:
- Figure 3 is too small and therefore difficult to read. The authors should present it in a better way or bigger, even if in the supplementary materials.
Response 6: Thanks for the comments. According to the comments, the demonstration of Figure 3 is changed, in order to improve the resolution. The (3A) mcr phylogenetic tree, (3B) nucleotide mutations and (3C) amino acid mutations of new mcr subtypes are displayed separately.
Point 7: Discussion:
- the discussion should be rewritten given that the authors repeat themselves throughout this section;
Response 7: Thanks for the comments. According to the discussion, we revised and rewrote the whole discussion section, and deleted redundant content, in order to better convey intended meanings. Four paragraphs of this section are organized as follows.
1) The resistant gene rates are related to the use of antibiotics of different sources, higher in livestock and poultry, and lower in wildlife.
2) The resistant gene rates of livestock and poultry, and human can be used for antibiotic usage monitoring.
3) Livestock and poultry should be the focus of antibiotic surveillance, as co-existence of mcr and blaNDM were found within single specimens, which possibly transmitted polymyxin and carbapenem resistance to humans simultaneously.
4) Summary and Limitation.
Point 8: Discussion:
- on line 201, the authors say that polymyxin was banned as an animal growth promoter in 2016, but in the Introduction is 2017;
Response 8: Thanks for the comments. China issued a ban of polymyxin use as an animal growth promoter in 2016, which was executed in 2017. The year information has been harmonized according to the comments as “The ban of polymyxin use as an animal growth promoter in 2017 seemed to…” on line 200.
Point 9: Discussion:
- line 202 and 203, abbreviations for these two organisms was already presented in the Introduction;
Response 9: Thanks for the comments. In accordance with the comments, the full names of CREC and MCRPEC were deleted from line 201.
Point 10: Discussion:
- Lines 203 - 205: how do the authors justify this fact?
Response 10: Thanks for the comments. In fact, we cannot justify the existence of polymyxin resistance in livestock and poultry collected in 2019, due to the limitation of gene-based (uncultured) strategy of this study. The possibility is speculated based on the finding of this study and our previous culture-based study (Table 2, 10.3389/fmicb.2020.00121), which showed colistin resistant strains being successively isolated from 2017 to 2019. To better demonstrate intended meaning, it has been revised as “However, the mcr detection rates in this study peaked in 2019 (10.60%, 167/1575) (Figure 2), in consistence with mcr-1 isolates successively recovered from 2017 to 2019, it speculated the possibility that polymyxin resistance still exist in livestock and poultry.” on line 201 to 204.
Point 11: Discussion:
- lines 205 to 210: given that supposedly the animals are not in contact with carbapenem, how do the authors explain the detection of these genes in the animals?
Response 11: Thanks for the comments. Sorry for the misleading words using previously. Carbapenem is not approved for food additives of livestock in China, but it may be used as drug treating animal diseases. Secondly, its use in humans pollute the environment and indirectly expose to animals. The last but not the least, some baseline bacteria with intrinsic blaNDM gene exist in animals. Accordingly, it has been revised on line 207 to 209.
Point 12: Materials and Methods
- this section has to be revised and more information and details provided prior to publication. There is no information on the type of samples collected, what were the protocols for sampling collection, transportation and every other necessary info. The authors state that they used genomic extraction kits from a certain manufacturer, but do not state which one, volume/amount of sample analysed, any concentration or pre-treatment performed to the samples, if using different kits, if they may have selected for one thing or the other. No information is provided about the sequencing of the samples, how was it done and performed, no information on the PCR conditions used (including the concentration of primers used, the mix used, the cycling conditions, although providing the annealing temperature in the supplementary information) and any controls used in this study
Response 12: Thanks for the comments. Gut specimen predominated the specimen types of this study, including human feces, animal anal swab, feces, intestinal content/swab or oral-pharyngeal swab, and poultry environment specimens related to gut environment, including drinking water, cage swab, depilator swab, cleaning sewage, chopping board swab, and soil. Unified protocols for specimen collection, transportation, and process were applied by professionals from local CDCs (Center for Disease Control and Prevention), hospitals, etc. Specimens were collected and transported in Cary-Blair Transport Medium, pre-treated and extracted nucleic acids by using 1.6 mL of each specimen and genomic extraction kit (TIANamp Bacteria DNA Kit, China). The nucleic acids samples were frozen for storage. The original amplification of mcr-1 to mcr-5 involved multiplex PCR, but single PCR was conducted with in this study. The number of PCR cycles for gene screening is 25 to 30, for CDS amplification it is 30. The PCR was performed using a 20μl volume containing 10μl Premix Taq version 2.0 (Takara, China), 8μl ultrapure distilled water, 0.5 μl (10μM) of each forward and reverse primer and 1 μl of DNA template. The amplified products were detected using gel electrophoresis and sequenced in both directions by Applied Biosystems 3730xl DNA Analyzer (Tsingke Biological Technology, Beijing). Accordingly, the above revisions have been added to Materials and Methods on line 263 to 284.

Reviewer 2 Report
This manuscript study how the characteristic of blaNDM and mcr Gene Distribution in Specimens from Different Regions of China.
The laboratory methods, result and discussion appear appropriate. The discussion section is also very well written. The total number of samples collected is appropriate. However, the authors should be specific what kinds of samples collected.
Major comments.
Comment 1. The methods section lacks some specific details about the sample collection methods, kind of the samples, gene sequencing and the PCR methods.
Comment 2. It would be interesting if the authors tried to include a correlation between the Microbiota profiles as well as the antibiotic resistance profiles. This will be very helpful in explaining the relationship that the authors attempt to clarify.
Minor comments
Comment 1. specify what type of specimens in the title.
Comment 2 similar issue in line 14 what kind of specimens.
Comment 3. Line 18, 19. This sentence need to rephrased. There is redundant here. You already mentioned that the positive rate of mcr is higher than the positive rate of blaNDM in the previous sentence.
Comment 4. Line 20 what do you mean by allelic variation. Did you detect several different copies of the gene?
Comment5. Figure 3 resolution is not readable. .
Acknowledging these shortcomings, the reviewer believes the manuscript would be suitable for publications.
Author Response
Response to Reviewer 2 Comments
This manuscript study how the characteristic of blaNDM and mcr Gene Distribution in Specimens from Different Regions of China.
The laboratory methods, result and discussion appear appropriate. The discussion section is also very well written. The total number of samples collected is appropriate. However, the authors should be specific what kinds of samples collected.
Response: Thanks for the reviewer’s comments and recognition of the study. We have studied your suggestions carefully and made revisions accordingly. The specific samples have been provided in details according to your comments (point 3 and point 4) in the revised manuscript.
Point 1: The methods section lacks some specific details about the sample collection methods, kind of the samples, gene sequencing and the PCR methods.
Response 1: Thanks for the comments. According to the comments, specific details about the sample collection methods and type of samples, gene sequencing and the PCR methods were added to 4.1 and 4.2 of method sections respectively:
“Gut specimen types of this study included human feces, animal anal swab, feces, intestinal content/swab or oral-pharyngeal swab, and poultry environment specimens related to gut environment, including drinking water, cage swab, depilator swab, cleaning sewage, chopping board swab, and soil. Unified protocols for specimen collection, transportation, and process were applied by professionals from local CDCs (Center for Disease Control and Prevention), hospitals, etc. Specimens were collected and transported in Cary-Blair Transport Medium, processed and extracted nucleic acids using genomic extraction kit (TIANamp Bacteria DNA Kit, China). The nucleic acids samples were frozen for storage.” on line 263 to 271.
“The PCR was performed using a 20μl volume containing 10μl Premix Taq version 2.0 (Takara, China), 8μl ultrapure distilled water, 0.5 μl (10μM) of each forward and reverse primer and 1 μl of DNA template. The amplified products were detected using gel electrophoresis and sequenced in both directions by Applied Biosystems 3730xl DNA Analyzer (Tsingke Biological Technology, Beijing).” on line 280 to 284.
Point 2: It would be interesting if the authors tried to include a correlation between the Microbiota profiles as well as the antibiotic resistance profiles. This will be very helpful in explaining the relationship that the authors attempt to clarify.
Response 2: Thanks for the comments. Due to the limitation of gene-based strategy, the profiles of microbiota as well as the antibiotic resistance are not available for this study. To better explain the relationship that we attempt to clarify, relative study was cited to discuss with the result of this study: “Compared with other sources, none polymyxin- or carbapenem-resistant strains [20], lower rates of mcr and blaNDM genes (Table 1) and less mcr genotypes (Table 2) are found in wildlife, which supports the hypothesis that wild animal is a net sink rather than a source of clinically relevant drug resistance [32]. The phenotypic diversity of drug re-sistant strains in wildlife is also low [32]. Wild animals are less likely to be exposed to antibiotics, the emergence of resistance genes possibly reflects the intrinsic resistance genes carried by some baseline bacteria.” on line 177 to 184.
Point 3: specify what type of specimens in the title.
Response 3: Thanks for the comments. According to the comments, the title has been revised to “blaNDM and mcr-1 to mcr-5 Gene Distribution Characteristics in Gut Specimens from Different Regions of China”. Gut specimens could represent the specimen type of this study, which included human feces, animal anal swab, feces, intestinal content/swab or oral-pharyngeal swab, and poultry environment specimens related to gut environment, including drinking water, cage swab, depilator swab, cleaning sewage, chopping board swab, and soil.
Point 4: similar issue in line 14 what kind of specimens.
Response 4: Thanks for the comments. According to Point 3 and Response 3, it has been specified as “gut specimens” on line 14.
Point 5: Line 18, 19. This sentence need to rephrased. There is redundant here. You already mentioned that the positive rate of mcr is higher than the positive rate of blaNDM in the previous sentence.
Response 5: Thanks for your comments. Sorry for the redundant sentence here, accordingly “The detection rate of blaNDM was much lower than mcr” has been deleted on line 19.
Point 6: Line 20 what do you mean by allelic variation. Did you detect several different copies of the gene?
Response 6: Thanks for your comments. We did not detect several different copies of the gene. The variation here means different gene subtypes. It has been revised as “multiple mcr genotypes detected in livestock and poultry.” on line 20 to 21.
Point 7: Figure 3 resolution is not readable.
Response 7: Thanks for your comments. According to your comments, the demonstration of Figure 3 is changed, in order to improve the resolution. The (3A) mcr phylogenetic tree, (3B) nucleotide mutations and (3C) amino acid mutations of new mcr subtypes are displayed separately.

Reviewer 3 Report
Lv and his/her colleagues (antibiotics-1103001) provides baseline information regarding the prevalence of blaNDM and some mcr genes in samplings from different parts of China. The provided data make added value for understanding the overall situation for the Chinese farming industry and human clinics etc...
In terms of a large amount of work for this investigation, the overall way for data presentation and logical results interpretation should be improved.
Title:
be accurate, only mcr1-5 was investigated.
Abstract:
Line 21-22. be accurate, what does "nearly all" mean?
what about blaNDM and mcr1-5 co-occurrence? please add this infor.
Introduction:
- be clear, not all polymyxin-resistant bacteria carry mcr gene!
- why blaNDM and mcr was studied, the introduction should be clearly documented, why these genes matter?
Results
Line 70, please use P < 0.05.
The quality of Figures 1 - 3 should be improved (too small to see), what colors for figure 1 are confusing, please group certain samples together and simplify the figure.
All figure legends are needed. What does new type mean? please use figure legend to detail them.
Discussion:
A limitation part should be added.
1) no bacterial culture and sampling, the genetic background information is not available.
2) limitation of PCR method, which based on the available primers.
3) limitation for searching for the new variants. It is not known how PCR and sequencing are conducted.
4) potential antimicrobial usage, colistin, and others, play a role in the enrichment of antimicrobial resistance genes in the gut or specimen samples/
5) Phylogenetic information or sequence diversity for blaNDM is also needed.
Methodology:
Line255-256. not clear? how many for each region and time period? how the samples are stored? a geography map for the samplings should be added as a figure.
Line265. There are ten mcr genes available and the detection method is available (10.1016/j.jhin.2021.01.010), however, only a partial mcr (1-5) was studied. Please make add an additional test for mcr-6 to mcr-10, or make a precise title. Please update this information with more recent literature.
References:
The overall references provided in this manuscript is very old, just an example.
The burden of mcr and baseline information should be cited:
10.1080/22221751.2020.1754133
10.3390/microorganisms7100461
The mechanism for the co-resistance should be included in the discussion
10.1128/mSystems.00783-20
Persistent infection for certain pathogens in China:
10.1128/mSphere.00163-20
An extensive literature study should put more recent findings in Chinese investigations together. A sufficient discussion in the context of this data is useful and comparable.
Author Response
Response to Reviewer 3 Comments
Point 1: Lv and his/her colleagues (antibiotics-1103001) provides baseline information regarding the prevalence of blaNDM and some mcr genes in samplings from different parts of China. The provided data make added value for understanding the overall situation for the Chinese farming industry and human clinics etc...In terms of a large amount of work for this investigation, the overall way for data presentation and logical results interpretation should be improved.
Response 1: Thanks for the reviewer’s comments. Your careful work and professional suggestions are highly appreciated. We have studied them carefully and revised accordingly, which present our work in a better way for data presentation and logical results interpretation.
Point 2: Title:
be accurate, only mcr1-5 was investigated.
Response 2: Thanks for the comments. According to the comments, the title has been revised to “blaNDM and mcr-1 to mcr-5 Gene Distribution Characteristics in Gut Specimens from Different Regions of China” on line 2.
Point 3: Abstract:
Line 21-22. be accurate, what does "nearly all" mean?
Response 3: Thanks for the comments. Sorry for not expressing accurately. It has been revised as “All the coexistence of mcr-1 and blaNDM gene and the coexistence of mcr genotypes within single specimen, and most new mcr subtypes was coming from livestock and poultry, and poultry environment.” on line 21 to 23.
Point 4: Abstract:
what about blaNDM and mcr1-5 co-occurrence? please add this infor.
Response 4: Thanks for the comments. In this study, all the co-occurrence of blaNDM and mcr were derived from blaNDM and mcr-1. The co-occurrence of mcr-2 to mcr-5 and blaNDM is not found. Accordingly, it has been clearly written as “coexistence of mcr-1 and blaNDM gene” in the abstract (line 21) and the rest of the manuscript.
Point 5: Introduction:
be clear, not all polymyxin-resistant bacteria carry mcr gene!
Response 5: Thanks for the reviewer’s comments. We completely agree with the comment. Sorry for the misleading description. It has been revised as, “Our previous work has shown, most of the mcr or blaNDM positive strains, which conferring polymyxin- and carbapenem-resistance, were normal flora.” on line 50 to 51.
Point 6: Introduction:
why blaNDM and mcr was studied, the introduction should be clearly documented, why these genes matter?
Response 6: Thanks for the comments. Genes blaNDM and mcr was studied for their encoding product New Delhi metallo-β-lactamase and phosphoethanolamine--lipid A transferase, which confer resistance for polymyxin and carbapenem respectively. These two antibiotics are among the last-resort antibiotics for defending against Gram-negative bacterial infections. As the reviewer addressed, many important mechanisms and genes are responsible for polymyxin or carbapenem resistance. Considering limited genes can be screened for 6,991 specimens, only mcr and blaNDM were chosen for their wide dissemination across species and regions.
It has been revised as, “Among the many mechanisms, blaNDM (New Delhi metallo-β-lactamase) gene and the mcr (mobile colistin resistance) gene conferring resistance to carbapenem and polymyxin respectively, exhibit cross-species and cross-region transmission [2].” on line 31 to 34.
Point 7: Results:
Line 70, please use P < 0.05.
Response 7: Thanks for the comments. P<0.05 has been used instead. It has been revised on line 69.
Point 8: Results:
The quality of Figures 1 - 3 should be improved (too small to see), what colors for figure 1 are confusing, please group certain samples together and simplify the figure.
Response 8: Thanks for your comments. According to the comments, Figure 1 has been revised, specimens belonging to the same source is simplified as same colour system. Animals and related specimens were adjusted to cool colour (livestock and poultry: blue; poultry environment: purple; wild animal: green), and human specimens to warm colour (yellow). The demonstration of Figure 3 has also been changed, in order to improve the quality shown. The (3A) mcr phylogenetic tree, (3B) nucleotide mutations and (3C) amino acid mutations of new mcr subtypes are displayed separately.
Point 9: Results:
All figure legends are needed. What does new type mean? please use figure legend to detail them.
Response 9: Thanks for your comments. The asterisk means certain new subtypes of mcr gene was found in certain year. Take the asterisk of 2010 as an example, it means new subtypes of mcr-3 was found in 2010. According to the comments, figure legend has been added to detail them as “* new subtypes of certain mcr genotype found in certain year” within Figure 2.
Point 10: Discussion:
A limitation part should be added.
1) no bacterial culture and sampling, the genetic background information is not available.
2) limitation of PCR method, which based on the available primers.
3) limitation for searching for the new variants. It is not known how PCR and sequencing are conducted.
4) potential antimicrobial usage, colistin, and others, play a role in the enrichment of antimicrobial resistance genes in the gut or specimen samples
5) Phylogenetic information or sequence diversity for blaNDM is also needed.
Response 10: Thanks for the comments. We are sorry for the in sufficient discussion of the limitations in this study. We deeply agree with the limitations addressed by the reviewer, each one of them is responded point to point and added to the last paragraph of discussion on line 239 to 249.
1) Due to the limitation of PCR method, bacterial culture or genetic information is not available in this study. It has been added as “On the other hand, bacterial culture and genetic background information is not available through gene-based strategy.” on line 243 to 245. The sampling information of specimens has been added as Figure S1 according to Point 11.
2) Due to PCR method, searching for the new variants is limited. According to the result of our study, mcr genotypes is not found out of the range it designed to amplify. For the same blaNDM primers used in this study and previous study, we found blaNDM-1, blaNDM-3, blaNDM-5 and blaNDM-24. It has been added to limitations as “Also, searching for the new variants is limited by PCR method. Although this method is improving through times [33,41], it is based on known genotypes data which often cannot discover unknown variant.” on line 247 to 249.
3) Limitation for searching for the new variants has been added as 2). The PCR conditions and sequencing method has been added as “The PCR was performed using a 20μl volume containing 10μl Premix Taq version 2.0 (Takara, China), 8μl ultrapure distilled water, 0.5 μl (10μM) of each forward and reverse primer and 1 μl of DNA template. The amplified products were detected using gel electrophoresis and sequenced in both directions by Applied Biosystems 3730xl DNA Analyzer (Tsingke Biological Technology, Beijing).” on line 280 to 284.
4) We totally agree that potential antimicrobial usage, colistin, and others, play a role in the enrichment of antimicrobial resistance genes in the gut specimens, which need further support of culture-based data. Hence, the result of our culture-based study has been compared with this study in discussion. It shows consistence of more mcr than blaNDM gene rates (line 169 to 170), and consistence between the mcr detection rates of specimens and isolation rates of polymyxin-resistant strains in livestock and poultry, diarrhea patients and wild animals (line 172 to 173). The limitation has been added as “Potential antimicrobial usage, colistin, and others, play a role in the enrichment of antimicrobial resistance genes in the gut specimens, which need further support of culture-based data.” on line 239 to 241.
5) According to the comments, the sequence diversity for blaNDM is added to the result 2.2 as, “The amino acid (aa) identity between NDM-1 and NDM-24 was 99.8%.” on line 112.
Point 11: Methodology:
Line255-256. not clear? how many for each region and time period? how the samples are stored? a geography map for the samplings should be added as a figure.
Response 11: Thanks for the comments. According to your comments, a geography map for the samplings was added as Figure S1, in order to demonstrate the specimens collected for each region and time period. The legends have been added as “Figure S1. Geographic distribution and frequency of the specimens collected from 2010 to 2020. The colour shades represent different years and the pie area reflects the number of samples collected in each region.” on line 308 to 310.
Specimens were collected and transported in Cary-Blair Transport Medium to laboratory according to unified protocol. Nucleic acids samples were extracted using genomic extraction kit (TIANamp Bacteria DNA Kit, China) and frozen for storage. Accordingly, it has been revised as on line 269 to 270.
Point 12: Methodology:
Line265. There are ten mcr genes available and the detection method is available (10.1016/j.jhin.2021.01.010), however, only a partial mcr (1-5) was studied. Please make add an additional test for mcr-6 to mcr-10, or make a precise title. Please update this information with more recent literature.
Response 12: Thanks for the comments and the ten mcr genes method provided. The information has been updated in the limitation of this study according to Point 10 as follows, “Although this method is improving through times [33,41], it is based on known genotypes data which often cannot discover unknown variant.” on line 248 to 249.
In accordance to your comments, the sub-title of the method has been specified as “blaNDM and mcr-1 to mcr-5 screening of gut specimens and sequence analysis” on line 272.
Point 13: References:
The overall references provided in this manuscript is very old, just an example.
The burden of mcr and baseline information should be cited:
10.1080/22221751.2020.1754133
10.3390/microorganisms7100461
The mechanism for the co-resistance should be included in the discussion:
10.1128/mSystems.00783-20
Persistent infection for certain pathogens in China:
10.1128/mSphere.00163-20
An extensive literature study should put more recent findings in Chinese investigations together. A sufficient discussion in the context of this data is useful and comparable.
Response 13: Thanks for the comments and references provided. We have studied them carefully and cited them in sections of introduction and discussion.
- Colistin and its Role in the Era of Antibiotic Resistance: An extended review (2000-2019) (1080/22221751.2020.1754133) has been cited as, “blaNDM (New Delhi metallo-β-lactamase) gene and the mcr (mobile colistin resistance) gene conferring resistance to carbapenem and polymyxin respectively, exhibit cross-species and cross-region transmission [2]” on line 34.
- Global Burden of Colistin‐Resistant Bacteria: Mobilized Colistin Resistance Genes Study (1980–2018) (3390/microorganisms7100461) has been cited as, “These strains sourcing from livestock and poultry, patients, and wildlife are mainly non-pathogenic organism, which is consistent with the finding of 47 countries across six continents in mcr-positive strain [31]” on line 164 and “Polymyxins are often used as therapeutic drugs and feed additives for animals, and they are used more for farmed animals in China, where the highest number of mcr-positive strains were reported[31]” on line 195 respectively.
- Emerging Transcriptional and Genomic Mechanisms Mediating Carbapenem and Polymyxin Resistance in Enterobacteriaceae: a Systematic Review of Current Reports (10.1128/mSystems.00783-20) has been cited as, “Recently, mcr-1 and blaNDM coexistence have also been reported in the United States, Venezuela, and Japan [37-39], which reduces treatment options for multidrug-resistant bacteria infections and increases the incidence and mortality of the infections [40].” on line 230.
- Persistent Asymptomatic Human Infections by Salmonella enterica Serovar Newport in China (10.1128/mSphere.00163-20) has been cited as, “Similarly, Salmonella enterica isolated from diarrhea patients and asymptomatic individuals showed equal carring of mcr, suggesting baseline bacteria as mcr carrier [33]” on line 186.
Response to English language and style:
Thanks for your suggestion. We are deeply sorry for the language problem caused by our writing and we have tried to fix them during revision. We have also invited professor Xuejie Yu for language editing this time, we have been in good partnership all along. Professor Yu is a Chinese American, who is the editors of many English-language journals. One of his latest publications in MDPI is: Pathogens. 2021 Jan 23;10(2):115. doi: 10.3390/pathogens10020115.

Round 2
Reviewer 3 Report
1. The figure legend is needed for figure S1, also the size of the pie chart needs a scale if it indicates the number of samplings.
2. Again, the quality of figure 1-3 must be improved. Some figures (1 and 3B) are impossible to be visualized. Either separate them into different ones or make them larger. The resolution must be improved (DPI > 300). Figure 2 looks also ugly, without a clear line to measure the bar. Finally, the bold and bigger text should be in a clear format.
Otherwise, this paper should also double-check the overall text.
Author Response
Response to Reviewer 3 Comments
Point 1: The figure legend is needed for figure S1, also the size of the pie chart needs a scale if it indicates the number of samplings.
Response 1: Thanks for the reviewer’s comments. According to the comments, a scale has been added to indicate the number of samplings at the bottom of figure S1. In addition, the full name of each region is listed in the figure legend as, “Figure S1. Geographic distribution and frequency of the specimens collected from 2010 to 2020. The colour shades represent different years and the pie area reflects the number of samples collected in each region. Abbreviation: BJ=Beijing, AH=Anhui, GS=Gansu, YN=Yunnan, GX=Guangxi, GZ=Guizhou, NX=Ningxia, IM=Inner Mongolia, QH=Qinghai, ZJ=Zhejiang.” on line 314 to 316.
Point 2: Again, the quality of figure 1-3 must be improved. Some figures (1 and 3B) are impossible to be visualized. Either separate them into different ones or make them larger. The resolution must be improved (DPI > 300). Figure 2 looks also ugly, without a clear line to measure the bar. Finally, the bold and bigger text should be in a clear format.
Response 2: Thanks for the reviewer’s comments. According to the comments, the quality of figure 1-3 has been improved during this revision by making each of them larger in the manuscript. The resolution is set to DPI 600 for all figures. The text within all figures has been in bold and bigger format.
We are sorry for not noticing Y axis missed in figure 2, and it is revised to be seen accordingly. The colour of bar has also been adjusted in figure 2.
To better demonstrate Figure 3B, the font size has been revised larger. As a result, Figure 3 cannot be displayed within same page of the manuscript. Therefore, panel 3A was shown in one page 5, panels 3B and 3C were shown on page 6.
Point 3: Otherwise, this paper should also double-check the overall text.
Response 3: We have double checked through the text for language and spell, and revised accordingly:
“in order to provide” has been revised to “providing” on line 57;
“that” has been added on line 162;
“are less likely to be exposed to antibiotics” has been revised to “have less chance of being exposed to antibiotics” on line 186 to 187;
“and” has been revised to “or” on line 194;
“reflect” has been revised to “reflects” on line 194;
“proposed” has been revised to “proposing” on line 241;
“which” has been added on line 242.
